# The influence of permafrost and other environmental controls on stream thermal sensitivity across Yukon, Canada

Andras J Szeitz[1], Sean K Carey[1]

[1]School of Earth, Environment and Society, McMaster University, Hamilton, Ontario, L8S 4L8, Canada

*Correspondence to*: Andras J. Szeitz (szeitza@mcmaster.ca)

**Abstract**

Thermal sensitivity, defined as the slope of a linear regression between stream and air temperature, is a useful indicator of the strength of coupling between atmospheric forcings and stream temperature, or conversely, of the presence of non-atmospheric thermal influences such as groundwater contributions to streamflow. Furthermore, thermal sensitivity is known to be responsive to environmental change. This study expands the current state of knowledge of stream thermal sensitivity in cold, northern regions across catchment scales, investigates the environmental controls of thermal sensitivity across a range of catchment dispositions, and assesses the thermal influence of environmental conditions unique to cold regions, namely permafrost. We conducted a linear regression analysis relating mean daily air and stream temperature in 57 catchments in Yukon, Canada, with catchment areas ranging from 5.4 to 86,500 km$^2$, and with catchment mean permafrost probabilities ranging from 0.0 to 0.99. Thermal sensitivities obtained from the linear regressions ranged from 0.14 to 0.84ºC ºC$^{-1}$, with a median of 0.56ºC ºC$^{-1}$, and the regression intercepts ranged from -0.07 to 7.60ºC, with the mean regression Nash-Sutcliffe efficiency = 0.81. Thermal sensitivity was positively related to catchment area, land covers representing surface water storage, and streamflow 'flashiness' or a lack of groundwater contributions. The greatest single environmental characteristic explaining the variance in thermal sensitivity was catchment topography and physiography (9% variance explained); however, 39% of the variance in thermal sensitivity was explained jointly by catchment physiography, land cover, and permafrost presence indicators, suggesting thermal sensitivity is the result of multiple interacting controls. Permafrost appeared to have indirect and offsetting effects on thermal sensitivity through its influence on separate and counter-acting processes controlling thermal sensitivity.

## 1. Introduction

Stream temperature is the master water quality variable as it mediates stream physical and biotic processes, and is a primary control on ecosystem productivity, aquatic species distributions, hydrochemistry through substrate weathering, and nutrient availability (Ebersole et al., 2001; Brown et al., 2004; Anderson, 2005; Caissie, 2006; McNamara et al., 2008; Parkinson et al., 2016; McDowell et al., 2017). Stream temperatures and thermal regimes are sensitive to changes in local or global environmental conditions, and northern latitudes have experienced rapid change in response to climate warming at twice the

global average rate (Vincent et al., 2015). Northern and cold regions have unique environmental conditions, such as permafrost presence, which strongly controls surface and subsurface hydrology, and which are also sensitive to climate warming (Walvoord et al., 2012; Woo, 2012). While there is increasing interest in understanding the influence of cold regions hydrological processes on stream temperature (e.g., King and Neilson, 2019; Sjöberg et al., 2021), there is a distinct lack of knowledge of the current state of northern stream thermal regimes and how they will respond to shifting climatic and hydrological conditions.

Stream temperature is ultimately controlled by the stream heat budget, and the primary processes governing stream temperature have received extensive attention over the past decades (Brown, 1969; Constantz, 1998; Webb and Zhang, 1999; Johnson, 2004; Moore et al., 2005b; Leach and Moore, 2010). Radiative heat exchanges (i.e., shortwave and longwave radiation) between the stream and the atmosphere often dominate the stream heat budget in all regions, including northern catchments alike, but local topography, channel morphology, and the timing and volume of streamflow and lateral inflows also influence stream temperature (Story et al., 2003; Caissie, 2006; King et al., 2016). Environmental controls through seasonal snow cover, local riparian vegetation and channel shading, permafrost presence and distribution, and wildfires (Isaak et al., 2010; Leach and Moore, 2010, 2015; MacDonald et al., 2014; Dugdale et al., 2018; King and Neilson, 2019; Wondzell et al., 2019; Sjöberg et al., 2021), as well as anthropogenic disturbances such as watercourse-impounding reservoirs, timber harvesting, and natural resource extraction (Bjerklie and LaPerriere, 1985; Lowney, 2000; Moore et al., 2005a) all exert additional controls on stream temperature by altering energy exchanges between the water and its surroundings. The processes governing stream temperature are complex, may act at the sub-reach or catchment scale, and often have offsetting influences. The cumulative effects of the atmospheric and environmental conditions controlling stream temperature ultimately describe how stream temperature responds to surrounding conditions.

Physically based stream temperature models can be used to investigate controls on stream temperature and enable robust predictions of its response to environmental change, but have historically been developed for site-scale studies and face ongoing limitations to regional implementation due to high input-data and computational demands (Dugdale et al., 2017). Linear regression analysis is a simpler alternate approach to investigate the environmental controls on stream thermal regimes by relating stream and air temperature (Crisp and Howson, 1982; Stefan and Preud'homme, 1993; Mohseni et al., 1998; Pilgrim et al., 1998). The regression slope, commonly termed the 'thermal sensitivity', provides insight into the degree of coupling between stream temperature and atmospheric forcings or the influence of non-atmospheric processes such as groundwater or glacier meltwater contributions to streamflow. Thermal sensitivity analysis has been effectively used in streams across a range of spatial and temporal scales, and can identify the environmental controls on thermal sensitivity and provide insight into how thermal sensitivities may respond to climate change (e.g., Ducharne, 2008; Vliet et al., 2011; Kelleher et al., 2012; Luce et al., 2014; Winfree et al., 2018; Leach and Moore, 2019; McGill et al., 2024). Most thermal sensitivity studies to date have focused on temperate regions and have developed into a robust body of literature. Recently,

however, there has been increasing interest in improving our understanding of stream thermal regimes in cold, northern regions, which have received far less attention but have unique hydrological processes that, when altered, will have uncertain consequences for stream thermal regimes (e.g., Lisi et al., 2015; King et al., 2016; Bolduc et al., 2018; Winfree et al., 2018; Docherty et al., 2019; King and Neilson, 2019; Fabris et al., 2020).

While many of the processes identified in temperate regions as controls of thermal sensitivity are likely to apply in northern regions as well, near-surface permafrost strongly alters surface and subsurface hydrology (Woo, 2012; Kurylyk and Walvoord, 2021), leading to uncertainty in the transferability of our current understanding of the controls of thermal sensitivity to northern regions. Permafrost presence affects hydrology through its influence on vegetation community composition and snowpack accumulation patterns, surface water bodies, and regional groundwater patterns (Jorgenson et al., 2001; Shur and Jorgenson, 2007; Pomeroy and Gray, 1995; Walvoord et al., 2012; Woo, 2012; Grünberg et al., 2020), all of which influence stream thermal sensitivity. Current research suggests that permafrost may have counteracting influences on stream temperature depending on scale and process, e.g.: by warming lateral inflows in headwater catchments or moderating the heat flux from hyporheic exchange (King and Neilson, 2019; Sjöberg et al., 2021), but the net effect of permafrost on thermal regimes across scales is unclear. Compounding our limited understanding of the role permafrost plays in influencing thermal sensitivity, northern environments have experienced rapid change in response to climate warming, with > 2ºC of warming already realized in northern Canada (Zipper et al., 2018; Bush and Lemmen, 2019; DeBeer et al., 2021), and permafrost degradation in response to climate change is expected to alter groundwater temperature and dynamics (Ge et al., 2011; Kurylyk et al., 2016; Chiasson-Poirier et al., 2020), with emergent flow pathways shifting the timing and magnitude of streamflow and therefore altering thermal sensitivity.

With the gaps in knowledge described above, we set our research objectives for this study to 1) describe the range and variability of stream thermal sensitivity across a range of catchment sizes and permafrost dispositions in Yukon, Canada; 2) identify key environmental controls on thermal sensitivity in northern streams; and 3) investigate the role that permafrost plays in influencing thermal sensitivity. This study also serves to set a 'baseline' of current stream thermal conditions against which future thermal regimes, in response to changing environmental conditions, may be compared.

## 2. Methods

### 2.1 Study area

Yukon Territory, in northwest Canada, has a southern border on 60ºN latitude and extends north to the Arctic Ocean, and has a total area of approximately 483,000 km$^2$ (Fig. 1). The physiography of Yukon varies from the maritime-influenced, glacierized St. Elias Mountains in the southwest, to tundra coast in the north, with a northward transition from boreal forest to taiga cordillera through the territory. Annual precipitation ranges from > 2,000 mm in the mountainous southwest to ~250

to 300 mm in lower elevations of the interior. Mean annual temperatures (MAT) range from -10ºC in the north to between -2 and 0ºC in the south. Interior Yukon has a continental climate which results in intense, convective precipitation events through the summer months. Much of Yukon has mean daily air temperature > 10ºC during the summer months of June, July, and August, bolstered by long daylight hours, and daily maximum temperatures can exceed 25ºC.

Areas with MAT consistently < 0ºC may be underlain by permafrost; and the permafrost extent is reflected in the MAT gradient from north to south (Fig. 1). Permafrost extents are classified as isolated, sporadic, discontinuous, or continuous, with permafrost coverage corresponding to < 10%, 10-50%, 50-90%, and > 90%, respectively. Permafrost presence is a strong control on surface characteristics (e.g., vegetation) and hydrology throughout Yukon.

## 2.2 Data

All data processing and analysis was conducted using the R programming language, version 4.2.0 (R Core Team, 2022).

### 2.2.1 Stream temperature and discharge

A data set of continuous sub-daily stream temperature (ºC) and discharge ($m^3$ $s^{-1}$) was acquired from multiple agencies that operate hydrometric stations in streams across Yukon. The Yukon Territorial Government Water Resources Branch (YTG-WRB) provided data for 15 streams it monitors; the Water Survey of Canada (WSC) provided data for 34 stations, and a further eight streams are monitored by the Watershed Hydrology Group (WHG, authors' affiliation) at McMaster University, for a total of 57 sites. Discharge time series were also requested, but were unavailable for seven of the streams. The sub-daily data provided by YTG-WRB and WHG were primarily at 15 minute intervals, while the WSC data were primarily at hourly intervals.

Data were omitted from the sub-daily data set if they met the following criteria: site-days where > 15% of the observations were missing; unrealistic water temperature (e.g., > 100$^o$C or <-60$^o$C); and quality flags indicating erroneous or potentially inaccurate data. Mean daily stream temperature and discharge were calculated from the filtered sub-daily data. Visual inspection of the stream temperature and discharge time series led to the omission of data from WSC station number 09BA001, as the data was of poor quality, and the omission of data from YTB-WRB station number 29AB009, as it was a partial duplicate of station number 29AB002. This study considered the post-freshet open water period, so data were further filtered to be within 1 July and 15 October, and stream and air temperature data < 0$^o$C were also omitted. The analysis-ready temperature and discharge data set had 35,597 values of mean daily stream temperature and 26,159 concurrent values of mean daily discharge spanning from 24 August 1997 to 30 August 2023.

The stream temperature measurements were obtained using a variety of instruments over their periods of record. Details regarding sensor accuracy and resolution for instruments known to have been used were compiled and assessed, but are

incomplete. Direct attribution of a sensor to all site-years of data is unfortunately not possible. Additionally, it should be noted that no information is available regarding the instruments used by the WSC. Given the uncertainty in the specific instrument used for any given data, it should be assumed that the lowest known sensor accuracy applies to all the data (i.e., accuracy of $\pm$ 0.44$^o$C).

### 2.2.2 Stream network and catchment delineations

Catchment delineations for the WSC stations were obtained from the National Hydro Network (NHN) Basin Polygon spatial data set (Water Survey of Canada, 2016). The delineations for the remaining stations operated by YTG-WRB and WHG were computed using the 32 m spatial resolution ArcticDEM v4.1 digital elevation model (DEM, Porter et al., 2023) as the input DEM for the catchment delineation processes available through the R package, *whitebox* (Wu and Brown, 2022). The catchment delineations' accuracy was assessed visually and small, manual corrections were made where required. Catchment areas ranged from 5.4 to 86,500 km$^2$.

Canada1Water has enhanced the NHN GeoBase data set by deriving stream network properties for all stream vectors included in the NHN (Canada1Water, 2023). The Strahler stream order (SO) and Shreve magnitude of each stream at the hydrometric station was extracted using this data set.

### 2.2.3 Meteorology

Due to the remote nature of many of the hydrometric stations and the sparse network of meteorological stations in Yukon, gridded climate reanalysis data were used to obtain consistent estimates of air temperature and other meteorological variables across all sites. ERA5-Land is a gridded climate data product that provides near-real-time global climate data at a spatial resolution of ~10 km at hourly or daily intervals (Muñoz-Sabater et al., 2021), and its greater accuracy in comparison to well-established but coarser resolution gridded data products (e.g., ERA5, NARR) has been demonstrated in recent stream temperature modelling research (Gatien et al., 2023; Mihalevich et al., 2022).

The ERA5-Land data set was queried using Google Earth Engine to extract continuous time series of the mean daily air temperature at 2 m above surface (ºC) and precipitation (mm), and daily catchment-averaged values of snow cover (as a proportion). Time series were extracted from 1991 to 2023, which permits the calculation of climate normals (e.g., 1991 to 2020). The air temperature and precipitation time series were extracted from the ERA5-Land grid cell overlying each hydrometric station location, and catchment mean snow cover values were extracted using the catchment delineations created as described in Sect. 2.2.2.

The ERA5-Land grid attributes a mean surface elevation to each grid cell. In mountainous regions, however, the earth surface will span a range of elevations within a given grid cell's domain. Because air temperature varies with elevation and

given that many of the hydrometric stations are located in mountainous regions, a temperature lapse rate adjustment was applied to the ERA5-Land air temperature time series as follows:

$$T_a = T_{a,E} + (z - z_E) \cdot \Gamma_m \quad (1)$$

where $T_a$ is the lapse-corrected air temperature (ºC), $T_{a,E}$ is the air temperature at 2 m above surface as extracted from ERA5-Land (ºC), $z$ is the elevation of a given hydrometric station (masl), and $z_E$ is the elevation (masl) of a given ERA5-Land grid cell that overlies a given hydrometric station, and $\Gamma_m$ is a month-specific adiabatic lapse rate adjustment (ºC $m^{-1}$) applicable to North America (Kunkel, 1989). Mean annual air temperature was calculated for each station from the lapse-adjusted air temperature time series.

Total daily precipitation was extracted for each hydrometric station and summed to total annual and annual summer (June, July, August) precipitation. Climate normals of total annual and summer precipitation were calculated for the period 1991 to 2020. The daily snow cover proportions were extracted for each catchment for the period of record, and the catchment mean and median daily values of snow cover were computed accordingly.

### 2.2.4 Landcover

Terrain indices were calculated and extracted for each catchment and catchment stream network using ArcticDEM v4.1 and the R package *terra* (Hijmans, 2023). Land cover classifications were extracted for each catchment from the ESA WorldCover 10 m 2020 v100 data set (Zanaga et al., 2021) and the fractional coverage of each land cover class within a catchment was computed. Catchment mean permafrost probability was derived from the Northern Hemisphere Ground Temperature Map (Obu et al., 2018), which is a gridded data set including estimated permafrost probability fraction at 1 km resolution developed using a 'temperature at the top of the permafrost' model representing permafrost conditions from 2000 to 2016, and catchment median active layer thickness (ALT) was extracted from a similar data set (Ran et al., 2021).

### 2.2.5 Indices and derived values

Shallow and regional hydrogeological patterns influence stream thermal regimes, where streams with relatively high groundwater contribution have lower thermal sensitivity (e.g., Kelleher et al., 2012; Wissler et al., 2022; McGill et al., 2024), and catchment soil storage and runoff response time is also hypothesized to influence stream temperature (Sjöberg et al., 2021). Given the lack of groundwater flux or soil storage data, several proxy indices were derived from the available streamflow data. These indices are the Baseflow index ($BFI$, dimensionless), Richards-Baker index ($RBI$, dimensionless), and the estimated coefficients from a streamflow recession regression.

The $BFI$ is an indicator of groundwater contributions to streamflow in the absence of local hydrogeological data (Gustard et al., 1992). $BFI$ is computed as the ratio of base flow to total discharge over a specified time period (e.g., annual or seasonal)

after a base flow separation algorithm is applied to the streamflow time series. Baseflow separation was conducted following the methods of Tallaksen and Lanen (2004) in the R package *lfstat* (Laaha and Koffler, 2022), and the long-term mean $BFI$ was computed for each stream over the study period (1 July to 15 October). Conversely to the $BFI$, the $RBI$ is an index of hydrograph 'flashiness' whereby rapid runoff generation indicates a lack of or unavailability of catchment storage (Baker et al., 2004). The $RBI$ is a simpler index that describes variation in flow response relative to total discharge; it was calculated for each station-year as follows:

$$RBI = \frac{\sum_{i=1}^{n} |Q_i - Q_{i-1}|}{\sum_{i=1}^{n} Q_i} \quad (2)$$

where $Q$ is mean daily streamflow (m$^3$ s$^{-1}$), and $i = 1$ on 1 July, and the long-term mean $RBI$ for each station was computed from the annual values.

The rate of streamflow recession can provide information about flow path complexity (Hinzman et al., 2020) as well as active layer thickness (i.e., soil storage, Sergeant et al., 2023) in catchments with substantial permafrost coverage. These catchment storage-discharge relationships were assessed by fitting a linear regression to the log-transformed recession curve as follows:

$$\ln\left(-\frac{dQ}{dt}\right) = \ln(\alpha) + \beta\ln(Q) \quad (3)$$

where $dQ/dt$ is the change in mean daily runoff per day (mm d$^{-2}$), and $\alpha$ (mm d$^{-2}$) and $\beta$ (-) are estimated coefficients.

### 2.3 Analysis on the controls of thermal sensitivity

The range of thermal sensitivity across Yukon streams, and the catchment controls on their magnitudes, were determined through a sequence of regression analyses. In general terms, the influence of groundwater contributions to streamflow (Johnson et al., 2020), runoff as influenced by catchment storage and permafrost disposition (Sjöberg et al., 2021), stream discharge and stream order (Webb et al., 2003; Kelleher et al., 2012), and the influence of catchment and channel topography on groundwater contributions (Hare et al., 2021) and stream shading (Rutherford et al., 1997) have been identified or suspected to be influences on stream thermal regimes. The suite of catchment characteristics and indices described in Sects. 2.2.4 and 2.2.5, and as provided in Table 1, were selected and assessed on the basis that they may explain regional controls in stream thermal sensitivity.

**Linear Regression**

Linear regression was used to determine site-specific relationships between air temperature and stream temperature and the influence of including discharge as a predictor in the linear regressions was assessed. A linear regression relating stream temperature to air temperature was applied to each stream:

$$T_w = TS \cdot T_a + BT + \epsilon \qquad (4)$$

where $T_w$ is mean daily stream temperature (ºC), $T_a$ is mean daily air temperature (ºC), $TS$ is the slope of the regression (ºC ºC$^{-1}$), $BT$ is the intercept (ºC), and $\epsilon$ is the residual error (ºC). The slope is commonly referred to as the stream's thermal sensitivity (TS), which represents the stream temperature response to a change in air temperature. Here, we refer to the intercept, $BT$, as the baseline temperature, which provides information regarding the thermal processes affecting a stream when air temperatures decrease to 0ºC. Equation (4) was expanded with the inclusion of discharge as a predictor variable to:

$$T_w = TS \cdot T_a + l \cdot Q + BT + \epsilon \qquad (5)$$

where $l$ is an estimated coefficient (ºC s m$^{-3}$).

A time trend analysis of estimated coefficients and residual errors was conducted by fitting (Eq. 4) to the pooled data set filtered to include only observations in the first two weeks of July. The resultant estimated coefficients were applied to weekly subsets of the full data set and the corresponding residuals, $\epsilon$, were calculated.

**Redundancy analysis, variance partitioning, and multiple regression**

The relationships between the estimated coefficients ($TS$ and $BT$) and catchment characteristics and indices were explored using redundancy analysis (RDA) and correlations to better understand the magnitude and direction of influence among the environmental variables and $TS$ and $BT$. RDA is a two-step process which involves 1) fitting multiple linear regressions between an explanatory variable matrix and a multivariate response matrix followed by 2) a principal component analysis of the fitted values. The RDA shows the correlations between multiple explanatory and response variables, and produces statistical models that can be tested for significance. Upon conducting an additional variance partitioning analysis, RDA provides estimates of the individual and combined proportions of variance explained by each predictor. Forward-selection of the full suite of catchment and environmental variables was conducted prior to the RDA to remove colinear predictors; Table 1 presents the forward-selected variables and variable grouping applied to the RDA. Additionally, Pearson's correlations were computed between TS and BT, and the forward-selected variables for catchments grouped by permafrost classification, to identify trends between the response and predictors across permafrost classes. Lastly, a stepwise multiple regression model was used to identify which catchment or environmental properties were significant predictors (assessed at $p < 0.05$) of $TS$.

The linear and multiple regressions' performance was assessed by calculating model-specific root-mean-squared error (RMSE, kg $m^{-3}$), and Nash-Sutcliffe efficiency (NSE, dimensionless) as follows:

$$RMSE = \sqrt{\frac{1}{m}\sum_{i=1}^{m}\left(\widehat{T_{w,i}} - T_{w,i}\right)^2} \quad (6)$$

$$NSE = 1 - \frac{\frac{1}{m}\sum_{i=1}^{m}\left(\widehat{T_{w,i}} - T_{w,i}\right)^2}{\frac{1}{m}\sum_{i=1}^{m}\left(T_{w,i} - \bar{T_{w,i}}\right)^2} \quad (7)$$

where $\widehat{T_{w,i}}$ is the estimated stream temperature (ºC), $\bar{T_{w_i}}$ is the mean stream temperature (ºC), and $m$ is the number of observations in the fitting data set.

## 3. Results

### 3.1 Temperature and discharge statistics

A broad range of meteorological and hydrological conditions were represented in the data set (Fig. 2). While the absolute range of $T_w$ was from 0.01ºC to 20.9ºC, the smaller streams (SO 2 and 3) had comparable median daily $T_w$ values of 4.9ºC and 4.5ºC, respectively, while the larger streams (SO 4 to 7) had median daily $T_w$ between 8.4ºC and 9.9ºC. The distribution of air temperature across the sites was similar, with a median $T_a$ ranging from 11.2ºC to 12.2ºC for sites with SO 3 to 7, while SO 2 and 8 had a median $T_a$ of 9.1ºC and 13.1ºC, respectively. Streamflow increased with SO, with median streamflows ranging from 0.12 to 85.5 $m^3\ s^{-1}$. Runoff was relatively uniform across all orders with median daily runoff values ranging from 0.74 to 1.31 mm $d^{-1}$, although smaller order streams appear to have greater runoff than the larger order streams.

Peak flows typically occurred during the snowmelt freshet in May, with 52% of the streams experiencing annual peak flow prior to 1 June; however, peak flows also occurred throughout the summer and are associated with intense convective storms. Stream temperature was strongly seasonal across all sites and permafrost classifications (Fig. 3), with streams warming from $T_w$ near 0ºC during freshet, to annual 7 day average $T_w$ maxima occurring through July and early August.

### 3.2 Thermal sensitivity

Site-specific RMSE and NSE from fitting Eq. (4) ranged from 0.46 to 2.65ºC, and 0.35 to 0.92, respectively, with corresponding means of 1.21ºC and 0.81. Catchments classified as continuous permafrost consistently had lower NSE than catchments with sporadic or isolated permafrost, which were consistently greater, while discontinuous permafrost catchments model performance increased with increasing stream order (Fig. 4). The mean NSE for each permafrost class was 0.73, 0.82, and 0.83 for continuous, discontinuous, and sporadic and isolated, respectively.

The range of $TS$ and $BT$ from fitting Eq. (4) was 0.14 to 0.84ºC ºC$^{-1}$ and -0.07 to 7.60ºC, respectively (Fig. 5). The minimum $TS$ observed was 0.14 for the glacierized Alsek River catchment; the low $TS$ of this catchment is attributable to its glacierized status, and due to the substantial thermal influence of glacial melt water. Values of $TS$ near 0 represent streams with low sensitivity to atmospheric conditions, while high values (e.g., $> 0.9$) indicate close coupling between $T_a$ and $T_w$. While the minimum $TS$ (0.14) corresponded to a glacierized catchment, eight other non-glacierized catchments had $TS < 0.3$ and 19 in total had $TS < 0.50$. Conversely, only six streams had a $TS > 0.70$, indicating few streams had high sensitivity. The $BT$ generally increased with decreasing permafrost coverage, with mean $BT$ increasing from 1.81 to 2.62ºC between continuous and sporadic and isolated permafrost classes and with increasing catchment area.

The addition of $Q$ as a predictor in Eq. (5) slightly improved model performance; RMSE and NSE ranged from 0.33 to 2.48ºC, and 0.54 to 0.93, respectively, with corresponding means of 1.14ºC and 0.84. The estimated coefficients representing $TS$ and $BT$ ranged from 0.08 to 0.86, and 0.25 to 7.58, respectively. Model statistics corresponding to Eqs. (4) and (5) are summarized in Table 2.

The addition of $Q$ as a predictor had a greater impact on the estimated $BT$ than the estimated $TS$. The difference in site-specific $TS$ estimated by the two models was an order of magnitude less than the fitted values; the $TS$ difference ranged from -0.06 to 0.08, with a $TS$ difference < 0.02 for 66% of the streams. In contrast, the difference in $BT$ estimated by both models was in the same order of magnitude as the estimated values, and ranged from -6.3 to 1.34. Equation (4) was re-fitted to the same subset of data as Eq. (5) and both models' Akaike Information Criterion (AIC) were compared to determine whether Eq. (5) was a significant improvement, using a threshold of $|\Delta AIC| > 2$. Equation 5 had an AIC 1357 less than Eq. (4) and was deemed a significantly better model.

Residual errors displayed increasing negative bias through the summer from the time trend analysis (Fig. 6). The mean residual error for each permafrost classification began decreasing by the second week of August; however, discontinuous permafrost catchments did not have an overall negative bias until the third week in August, and sporadic and isolated catchments until the beginning of September. Increasing negative bias indicates the fitted coefficients representing 1 July to 14 July increasingly over-estimated $T_w$ through the summer, with greater overestimation with increasing permafrost extent.

### 3.3 Redundancy analysis

#### Catchment physiography

Variance in $TS$ and $BT$ was generally consistently explained by catchment physiography metrics (Panel (a), Fig. 7). A greater portion of variance in $TS$ and $BT$ was explained by topographical variables (e.g., stream network $TPI$, catchment average slope and $TRI$) than the 'magnitude' variables of catchment area or Shreve stream magnitude. Topographical

variables were consistently correlated across permafrost classes (Fig. 8), and had stronger correlations with $TS$ (Pearson $r$ ranged from -0.89 to 0.84) than with $BT$, which was weakly correlated with the 'magnitude' and 'shading' metrics, with $r$ values of 0.37 and 0.36 for $\log_{10}(A)$ and $TRI$, respectively. $TS$ was negatively related to greater catchment slope and terrain shading ($TPI$ has the opposite relation because more positive values of $TPI$ indicate less topographic shading), whereas both 'magnitude' variables were positively correlated with $BT$.

**Climatology**

In comparison to catchment physiography, climate variables were generally weakly correlated to $TS$ and $BT$ (Panel (b), Fig. 7), and explained a smaller portion of variance. The greatest correlation between $TS$ and a climatological variable was with $MAT$, as $r$ ranged from -0.63 to 0.67 for continuous and discontinuous permafrost classes, respectively. Summer and total precipitation had a variable correlation with $TS$ across permafrost classes, with $r$ ranging from -0.52 for total precipitation in continuous permafrost, to 0.36 in discontinuous permafrost (Fig. 8). Longer seasonal snow cover was positively correlated with $BT$, but due to the sign of the relationship, this is suspected to relate to catchment size and the associated stream network heat accumulation (Panel (a), Fig. 7) rather than persistent snow packs suppressing $BT$ with cold melt water.

**Land cover**

Of the RDA models, land cover explained the most variance in $BT$ and $TS$, but there was substantial variability in the magnitude and direction of the correlations between the predictor and response variables (Panel (c), Fig. 7). The patterns of correlations were generally consistent across permafrost classes (Fig. 8), with the strongest correlations between thermal sensitivity and catchment tree cover ($r$ ranged from 0.40 to 0.46); this positive relationship likely reflects the probability of greater forest cover in larger catchments rather than a direct influence on $TS$. Land covers negatively related to $TS$ were moss and lichen, and bare land ($r$ ranged from -0.66 and -0.34, across both variables, respectively), but these land covers would not be expected to increase stream shading, and hence moderate $TS$. They were, however, correlated with catchment physiography variables that were negatively correlated with $TS$ (e.g., $r$ = 0.67 with mean catchment slope), and therefore likely overlap with catchment physiographical variables in the total variance explained. Across permafrost classes, $BT$ was consistently positively related to land covers representing surface water storage (i.e., water and herbaceous wetland) with the greatest correlations for continuous and sporadic and isolated permafrost catchments (Fig. 8), which, if connected to the stream network, can act as sources of relatively warm water during the summer.

**Permafrost indicators**

Variance in $TS$ and $BT$ was explained by a gradient in catchment permafrost disposition (e.g., mean permafrost probabilty and $ALT$) and in the catchment flow regime (i.e., whether streamflow is dominated by flashy flow response or baseflow contributions) as seen in Panel (d) of Fig. 7. The correlations between these 'permafrost indicator' variables and $TS$ were not consistent across permafrost classes, with continuous permafrost catchmetns often having opposing relationships with $TS$ in comparison to the other permafrost classes (Fig. 8). The greatest correlations in continuous permafrost catchments were between $TS$ and the flow regime variables, with $r$ = 0.87 and 0.80 for $\alpha$ and $RBI$, respectively, while $BFI$ had an $r$ = -0.88. Thermal sensitivity was positively correlated with catchment mean permafrost probability in continuous permafrost catchments ($r$ = 0.55), and negatively correlated in the other permafrost classes with $r$ = -0.57 and -0.26 for discontinuous and sporadic and isolated permafrost catchments, respectively. There is likely some collinearity in the variance explained by these predictors, as both permafrost probability and $ALT$ had moderate to strong correlations with the flow metrics. For example, the strongest correlation was between permafrost probability and $BFI$ with $r$ = -0.68, and the weakest correlation is between $ALT$ and $RBI$ with an $r$ = -0.52.

The influence of permafrost disposition in explaining variance in $BT$ and $TS$ was variable among the RDA models (see colour legend in Fig. 7). Baseline temperature appeared to have the strongest relationship with the gradient in permafrost disposition, with $BT$ generally increasing with decreasing permafrost coverage for all RDA models except catchment physiography. The overlap in permafrost classifications in the RDA, however, precludes a definitive relationship.

**Variance partitioning**

Variance partitioning revealed that all of the individual RDA models (Fig. 7) had statistically significant individual contributions to explaining the variance in $TS$ and $BT$ ($p$ < 0.05, Panel (a), Fig. 9) when accounting for the other RDA models. The individual RDA models explained between 9% to 14% of the variance in $TS$ and $BT$; the adjusted explanatory power of each RDA model was similar (ranged from 36% to 39%), except for the climatology RDA which only explained 15% of the total variance with the other models considered.

The individual RDA models had less explanatory power for $TS$ alone despite having the greatest combined adjusted $R^2$ of 0.73 (Panel (b), Fig. 9). Only the catchment physiography RDA model had a statistically significant individual contribution (grey oval, Panel (b), Fig. 9) explaining 9% of the total variance. The physiography and permafrost RDA models (grey and pink ovals, respectively, in Panel (b) Fig. 9) had the greatest explanatory power overall, explaining 76% and 52% of the variance in $TS$, but a substantial portion of each model's skill is redundant with the other models. A combined 39% of the variance in $TS$ was explained jointly by the catchment physiography, land cover, and permafrost indicator RDA models,

indicating that $TS$ is an integrated response to multiple catchment properties and is relatively insensitive to any one type of catchment characteristic.

Similarly to the full RDA model, the individual RDA models all were statistically significant in their individual contributions to explaining variance in $BT$ (Panel (c), Fig. 9). The land cover and climate RDA models were the greatest individual contributors with 33% and 28% of the variance in $BT$ explained, respectively. The permafrost RDA model explained 21% of the variance, while catchment physiography explained only 10% by itself. The combined adjusted $R^2$ of 0.67 was comparable to the full model ($R^2$ = 0.70) and to the $TS$ variance partitioning, but there was less redundancy or collinearity in how catchment properties explain variance in $BT$.

### 3.4 Multiple regression

The catchment properties used in the RDA (Table 1) and $TS$ coefficients from the linear model were used to fit a multiple linear regression model through stepwise model selection (retention of additional predictor variables assessed at $p < 0.05$). For this analysis, the $TS$ was modelled for the 46 catchments with streamflow records to permit inclusion of streamflow-derived candidate variables (e.g., $BFI$, $\alpha$). The final model form was as follows:

$$\widehat{TS} = 0.0947 \cdot \log_{10}(A) - 0.0168 \cdot CS + 0.0262 \cdot W + 0.457 \cdot \alpha + 0.428 \quad (8)$$

where $\widehat{TS}$ is the predicted thermal sensitivity (ºC $^{\circ}\mathrm{C}^{-1}$), $\log_{10}(A)$ is the log of catchment area, $CS$ is the mean catchment slope (º), $W$ is surface water land cover as a proportion of catchment area (-), $\alpha$ is the intercept of the recession curve (mm $\mathrm{d}^{-1}$ $\mathrm{d}^{-1}$), respectively. The model described by Eq. (8), as shown in Fig. 10, had an $R^2$ of 0.76, with standard error of 0.076ºC $^{\circ}\mathrm{C}^{-1}$, and a mean residual error of essentially zero (-1.1 × $10^{-18}$ ºC $^{\circ}\mathrm{C}^{-1}$). During model selection, one candidate model of note was identified, as it related the influence of catchment permafrost variables to $TS$. While this candidate model was not the best performing, it was fully statistically significant, and had the following form:

$$\widehat{TS} = -0.326 \cdot PF - 0.00429 \cdot ALT + 1.171 \quad (9)$$

where $PF$ is the catchment mean permafrost probability (-), and $ALT$ is the catchment median active layer thickness (cm). The $TS$ for one catchment (Blind Creek, a $3^{rd}$ order stream in a discontinuous permafrost catchment near Faro, Yukon) is not well estimated by the model, with a model error of 0.24 ºC $^{\circ}\mathrm{C}^{-1}$; there are no clear indications that explain this deviation, so the catchment was retained for the analysis. The regression in Fig. 10 shows that $TS$ is well predicted across all stream orders and all classifications of catchment permafrost disposition. A simpler model with only $\log_{10}(A)$ and mean catchment slope as predictor variables had slightly worse performance ($R^2$ = 0.73, standard error = 0.081 ºC $^{\circ}\mathrm{C}^{-1}$), but benefits from parsimony and ready application to ungauged basins; it has the following form:

$$\widehat{TS} = 0.0858 \cdot \log_{10}(A) - 0.0172 \cdot CS + 0.540 \quad (10)$$

A multiple regression model to estimate $BT$ performed notably worse in comparison to the model for $TS$. The best performing model, on the basis of $R^2$ and model standard error, had herbaceous wetland and surface water land covers as predictor variables with model statistics of $R^2$ = 0.49 and standard error = 1.14ºC. The model residuals, however, showed an increasing trend in model under-prediction and indicated the model did not fully represent the catchment properties that control $BT$, despite the inclusion of surface water storage predictor variables that can represent sources of warm water to a stream network and hence influence $BT$.

## 4. Discussion

### 4.1 Model performance and seasonal trends

Numerous other studies have demonstrated improved model skill by using a non-linear regression (e.g., Mohseni et al., 1998; Kelleher et al., 2012), functional regression methods such as generalized additive models (e.g., Laanaya et al., 2017; Boudreault et al., 2019), or sinusoidal functions (Johnson et al., 2020) when fitting models to time series of air-stream temperature data. Despite the improved model skill of these other techniques, linear regression remains a robust approach to relate the thermal sensitivity of a stream to its environmental and atmospheric forcings, and to investigate the controls on its variability especially when applied to seasonal subsets of data. As this study only considered the post-freshet open water season, the thermal sensitivity and intercept coefficients in Fig. 5 represent the air-stream temperature processes active during a period with minimal snowmelt influence, increasing active layer thickness, and includes peak water temperatures. These characterizations assist in investigating what influence permafrost, as a distinct trait of northern catchments, may have on stream thermal regimes.

Model performance with a simple linear model (Eq. 4) across the range of catchment areas and environmental conditions was good, with a mean NSE of 0.81. Catchment area (as proxied by SO) and permafrost disposition appears to influence the relationship between $T_w$ and $T_a$ (Fig. 4), with NSE generally increasing with SO and lower for continuous permafrost catchments in comparison to discontinuous or sporadic and isolated permafrost catchments. These differences in model skill suggest greater complexity in stream-air temperature relationships at smaller scales, and in catchments predominantly underlain by permafrost. Model performance was slightly improved through the addition of discharge as a predictor variable, with a mean NSE = 0.84 and a $\Delta$AIC > 2, but resulted in similar estimated $TS$ for 66% of the sites. As such, the application of a simple linear model relating $T_w$ and $T_a$ is likely sufficient for the purposes of investigating the environmental controls on $TS$.

The stream-air temperature relationship is dynamic over the study period (Fig. 6) across catchments of all permafrost classification, with increasing model over-prediction through August to October. However, the estimated coefficients fitted

to the pooled data set produced a distinct response for continuous permafrost catchments, with an earlier onset of model over-prediction beginning in early August, and a clear negative bias present by September. This contrasts the relatively synchronized timing and distribution of prediction errors for the other permafrost classes. The presence of temporal trends in time trend analysis residuals is consistent with previous research reporting on seasonal variability in $TS$ across the continental USA, but Segura et al. (2015) found the distribution of $TS$ for fall to be slightly less than summer, with greater differences in spring $TS$, and with greater regression intercepts from summer to fall (v. Fig. 12, Segura et al., 2015). While the residual time trend seen in Fig. 6 is consistent with literature, the relatively greater deviation by continuous permafrost catchments suggests they are subject to different controls on the stream-air temperature relationship in comparison to catchments with less permafrost presence.

### 4.2 Environmental controls of thermal sensitivity and baseline temperature

The thermal sensitivities reported here, with a range of 0.16 to 0.84ºC ºC$^{-1}$, and median of 0.56, are slightly lower than the ranges reported by studies focused on more temperate regions but are comparable if not greater than studies investigating summer $TS$ in southern Alaska (Lisi et al., 2015; Winfree et al., 2018), or in high elevation mountain catchments in northwestern North America (e.g., Wissler et al., 2022; McGill et al., 2024). Directly comparing stream thermal regime studies is complicated by the lack of standard approaches and methods (e.g., time periods considered, regression models applied, catchment areas included, etc.) among studies. However, when considering the underlying controls on thermal sensitivity, such as catchment area and slope, land cover, and groundwater contributions to discharge, the results presented here are consistent with the literature.

The primary controls on thermal sensitivity were related to catchment properties affecting stream water residence time, surface water storage, and subsurface runoff processes. The results indicating the dampening of $TS$ by steeper catchments, through greater flow velocity, and of increasing $TS$ with greater catchment area, due to greater flow volumes and longer exposure times between the stream and the atmosphere, are well supported by previous research (e.g., Donato, 2002; Isaak et al., 2012; Lisi et al., 2015; Ducharne, 2008; Kelleher et al., 2012; Segura et al., 2015). These catchment physiographical controls appear to have consistent influence regardless of study region.

Greater surface water storage, as represented through the land cover classes of ‘water’ and ‘herbaceous wetland’, had a positive effect on $TS$. In the context of northern summers with long daylight hours and with greater residence time compared to streams, surface water bodies may act as stores contributing relatively warm water to a stream network, increasing its $TS$. While this relationship has also been observed elsewhere (e.g., Mellina et al., 2002), greater lake coverage in southwest Alaska was found to have a dampening effect on $TS$ by Lisi and Schindler (2015), or to be a non-significant influence in southeast Alaska (Winfree et al., 2018). The effect of surface water bodies on $TS$ is likely influenced by local or regional

conditions that govern the thermal regimes of the surface waters; elevation, volume, sources of inflows, and topology may all influence whether lakes and surface waters buffer or bolster $TS$ in a given study region.

The $\alpha$ coefficient, representing subsurface runoff processes as a metric of streamflow flashiness, was positively related to $TS$. The influence of flashiness as represented by $\alpha$, or $RBI$ for that matter, has not been directly reported previously to the best of our knowledge. Some analogous comparisons may be made, however, as McGill et al. (2024) reported that upland catchments with thin soils overlying impermeable bedrock had greater $TS$ in comparison to catchments with greater soil depth. Segura et al. (2015) investigated the influence of subsurface flow contact time as a predictor of thermal sensitivity as well as the intercept. While Segura et al. (2015) did not find subsurface contact time to be a significant predictor of $TS$, it was negatively correlated with the intercept (i.e., $BT$). The suggestion of McGill et al. (2024) that thin soils which rapidly transmit infiltrated water to the channel, and have limited groundwater contributions, result in greater thermal sensitivity is consistent with this study and with the conceptual representation of $\alpha$ both as a metric of flashiness and a variable related to active layer thickness (Brutsaert, 2005; Sergeant et al., 2023). In systems with flashy flow regimes, a lack of thermal buffering from cold (relative to air temperature) groundwater contributions may be compounded by the greater thermal sensitivity associated with decreased flow volumes. While the variables representing flashy runoff generation most likely represent the absence of groundwater contributions to baseflow as a thermal sensitivity moderator, this study cannot conclusively describe the influence that rapid runoff generation in concert with subsurface heat exchange processes between infiltrated water and the soil has on thermal sensitivity, particularly in a permafrost context.

The regression intercepts, while less commonly reported in stream temperature studies, are relevant in the cold-regions as: 1) permafrost presence creates a cold interface, at maximum 0ºC, which has implications for the heat exchange processes between runoff-generating soil water and the frost table; and 2) catchments with near-continuous and continuous permafrost are commonly understood to have limited hydraulic connection to deep or regional groundwater and have streamflow regimes dominated by near-surface runoff generation processes. Greater permafrost presence appears to suppress the regression intercept, when considering the distribution of fitted intercept values (Fig. 5), and is consistent with the negative relationship seen between $BT$ and $PF$ in the RDA (Panel (d), Fig. 7). Streamflow indicators associated with greater permafrost presence, such as a low $BFI$, high $RBI$, or greater $\alpha$, all relate negatively with the regression intercept (Panel (d), Fig. 7 and Fig. 8) and provide further support for the presence of this process. When considered alongside the regression intercepts from other studies in the continental USA (e.g., Pilgrim et al., 1998; Wissler et al., 2022), the intercepts reported here are relatively low for the summer period, with a median value of 1.88ºC across all catchments.

The RDA results (Figs. 7 and 9) illustrate the complex relationship between $TS$ and $BT$ and their environmental controls. While there is redundancy among catchment physiographical variables, they are strong predictors of $TS$. Likewise, as permafrost presence strongly influences flow regimes and groundwater contributions, there is redundancy between those

predictors as well, but with greater importance for $BT$ than $TS$. Stream thermal sensitivity appears to be an emergent property of multiple aspects of the surrounding environment; the only variable class significantly explaining variance alone are topographical variables (9%, Panel (b) in Fig. 9), but can account for 63% of the variance in $TS$ together with the variable classes of land cover and permafrost indicators. This contrasts $BT$, where each variable class alone can significantly explain variance (Panel (c), Fig. 9), but with less predictive power, as has been reported in other studies investigating environmental controls on $BT$ (Segura et al., 2015; McGill et al., 2024). The inter-relatedness of the controls of thermal sensitivity suggest that environmental changes may result in a non-linear response of thermal sensitivity in northern catchments.

### 4.3 The role of permafrost and implications for stream temperature in a changing environment

The results of this study suggest that permafrost presence does not have a dominant control on the stream-air temperature relationship, but rather sets conditions for other catchment properties and processes that influence $TS$, some of which are counteracting. Extensive permafrost clearly suppresses groundwater contributions to baseflow, resulting in lower baseflow volumes which would be more responsive to atmospheric forcings and greater $TS$ (Panel (d), Fig. 7; Fig. 8; $\alpha$ coefficient in Eq. 8). Ice wedge polygon ponds, wetlands, and fens provide surface water storage in landscapes with extensive permafrost (Woo, 2012), and may increase both $TS$ and $BT$ through their contributions, at a greater relative temperature, to streamflow, as observed by Docherty et al. (2019) in their study of five headwater streams in north-eastern Greenland. However, the variable correlations ($r$ ranged from -0.56 to 0.55) and negative estimated regression coefficient (Eq. 9) between permafrost and TS suggest there are also moderating influences on $TS$ due to permafrost presence. Permafrost underlying stream channels acts as a significant heat sink through hyporheic exchange (King and Neilson, 2019), and is a process likely to moderate $TS$. Melting ground ice released through seasonal active layer expansion, contributing cool water to streamflow, or the heat sink of the frost table interacting with soil water (Woo, 2012), may also moderate $TS$. Aufeis, commonly found in northern regions with permafrost presence, although not inherently caused by permafrost (Turcotte et al., 2024), can act akin to in-stream snow banks in moderating summer stream temperatures as a source of cold melt water to the stream (Bolduc et al., 2018).

The relative contributions of these processes are likely variable and dependent on catchment-specific characteristics and properties and, together with the counteracting nature of permafrost's influence on $TS$, leads to uncertainty in response of $TS$ to permafrost degradation in response to climate change. Processes that presently increase $TS$ may shift to moderate $TS$ (e.g. greater groundwater contributions to streams as permafrost degrades, Walvoord et al., 2012), and vice versa (e.g., less ground ice forming annually under warmer conditions). As these permafrost-influenced processes do not act in isolation from other processes known to influence $TS$ and which are also expected to continue changing in the coming decades (e.g., precipitation timing and phase, riparian shrubification, air temperature), resolving the complexities of their interactions in

response to climate change is not trivial. There is substantial uncertainty in how northern stream thermal sensitivity, as an integrated catchment signal, will respond to climate change and further research into the individual controls on thermal sensitivity, and their responses to climate change, is needed.

## 5. Conclusion

The range and variability of summer stream thermal sensitivity observed in Yukon was comparable to values reported in temperate regions, although with lower maximum values across catchments of all permafrost classifications. The influence of cold, northern region hydrology on thermal sensitivity is apparent, with a median thermal sensitivity = 0.56ºC $ºC^{-1}$ across all catchments. The baseline temperatures (i.e., regression intercepts) were moderated by greater permafrost extent, although catchments of all permafrost disposition had numerous instances of low ($<$ 2ºC) baseline temperature.

Thermal sensitivity was suppressed by greater catchment slope, and was positively related to catchment area or stream order, surface water storage as represented by land cover classification, and streamflow 'flashiness'. The physiographical controls of in-stream residence time and heat accumulation are found to be applicable and dominant controls on thermal sensitivity in northern environments. Surface water bodies connected to stream networks may supply relative warm water through the summer season, bolstering thermal sensitivity and baseline temperature. Groundwater contributions represented by a baseflow index is commonly identified in temperate regions as a moderating control on thermal sensitivity, but flashy streamflow regimes are characteristic of continuous permafrost environments and indicate limited deep groundwater discharge to streams.

Permafrost appears to have offsetting influences on thermal sensitivity. The limiting influence of greater permafrost extent on baseflow contributions increases thermal sensitivity, but there is a simultaneous moderating influence of permafrost, possibly attributable to melting ground ice, lateral advection of hillslope runoff cooled by the frost table interface, in-stream aufeis, or heat loss to streambeds underlain by permafrost through hyporheic exchange processes and/or heat conduction. As permafrost degrades in response to climate change, we expect a dynamic response in the thermal regimes and thermal sensitivity of streams in environments with continuous and discontinuous permafrost. The net effect of these changes is uncertain due to the complex interactions between permafrost disposition, seasonal snow cover, precipitation timing, and subsurface hydrology. However, identifying current thermal sensitivities across catchment scales and permafrost conditions may help provide insight into the thermal evolution of streams in northern regions.

## Data Availability Statement

The data used in this study is available in a Zenodo repository (DOI: 10.5281/zenodo.11527471).

**Author Contributions**

AJS contributed to the research design, compiled the data set, conducted the data processing, analysis, and manuscript preparation. SKC contributed to the research design and manuscript preparation.

**Conflict of Interest**

The authors declare that they have no conflict of interest.

**Acknowledgements**

This research was funded by a Natural Sciences and Engineering Research Council (NSERC) Discovery Grant to S. Carey, and an NSERC CGS-D awarded to A. Szeitz. We would like to thank the Yukon Territorial Government – Water Resources Branch, and the Water Survey of Canada for providing streamflow and stream temperature data.

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

## Tables

770 Table 1: Environmental properties extracted for each catchment, or calculated at the catchment hydrometric station for streamflow-derived variables ($BFI$, $RBI$, $\alpha$). The variable groupings were used to conduct the redundancy analysis and variance partitioning.

| **Variable grouping** | **Variable** | **Symbol** | **Range** | **Unit** |
|---|---|---|---|---|
| Catchment physiography | Log of catchment area | log10($A$) | 0.74 - 4.11 | - |
| | Shreve stream magnitude | Shreve | 2 - 4724 | - |
| | Mean catchment slope | Ca slope | 4.5 - 25.2 | ° |
| | Mean catchment Terrain Ruggedness Index | Ca TRI | 1.96 - 12.2 | m |
| | Mean stream network Topographic Position Index | SN TPI | -3.41 - - 0.48 | m |
| Climatology | Mean annual temperature | MAT | -7.0 - 0.8 | °C |
| | Annual precipitation | Total ppt | 418 - 894 | mm |
| | Summer precipitation | Summer ppt | 166 - 391 | mm |
| | First snow-free day | 1st snow-free day | 156 - 211 | day of year |
| Land cover | Tree | tree | 0.0 - 95.5 | % |
| | Shrubland | shrubs | 0.0 - 35.6 | % |
| | Grassland | grass | 2.1 - 72.9 | % |
| | Cropland | crops | 0.0 - 0.03 | % |
| | Built up | built up | 0.0 - 0.13 | % |
| | Bare | bare | 0.0 - 27.1 | % |
| | Snow and ice | snow and ice | 0.0 - 10.0 | % |
| | Water | water | 0.0 - 5.5 | % |
| | Herbaceous wetland | herbs | 0.0 - 0.26 | % |
| | Mangroves | mangroves | 0.0 - 0.01 | % |
| | Moss | moss | 0.2 - 24.9 | % |
| Permafrost indicators | Intercept of the recession curve | α | 0.02 - 0.35 | mm $d^{-2}$ |
| | Baseflow Index | BFI | 0.15 - 0.96 | - |
| | Richards-Baker Index | RBI | 0.03 - 0.45 | - |
| | Catchment mean permafrost probability | PF | 0.00 - 0.99 | $proportion$ |

| **Variable grouping** | **Variable** | **Symbol** | **Range** | **Unit** |
|---|---|---|---|---|
| | Catchment median active layer thickness | ALT | 67.4 - 176 | cm |

Table 2: The range of estimated thermal sensitivity (TS) and baseline temperature (BT) coefficients for the two linear regressions (Eqs. 4 and 5). The root-mean-square error (RMSE) and Nash-Sutcliffe efficiency (NSE) are the associated model goodness-of-fit statistics.

| **Model** | **Range** | **Range** | **Range** | **Mean** | **Range** | **Mean** |
|---|---|---|---|---|---|---|
| Eqn. 4: $T_w = TS \cdot T_a + BT + \epsilon$ | 0.14 to 0.84 | -0.07 to 7.60 | 0.46 to 2.65 | 1.21 | 0.35 to 0.92 | 0.81 |
| Eqn. 5: $T_w = TS \cdot T_a + l \cdot Q + BT + \epsilon$ | 0.08 to 0.86 | 0.25 to 7.58 | 0.33 to 2.48 | 1.14 | 0.54 to 0.93 | 0.84 |

780 **Figures**

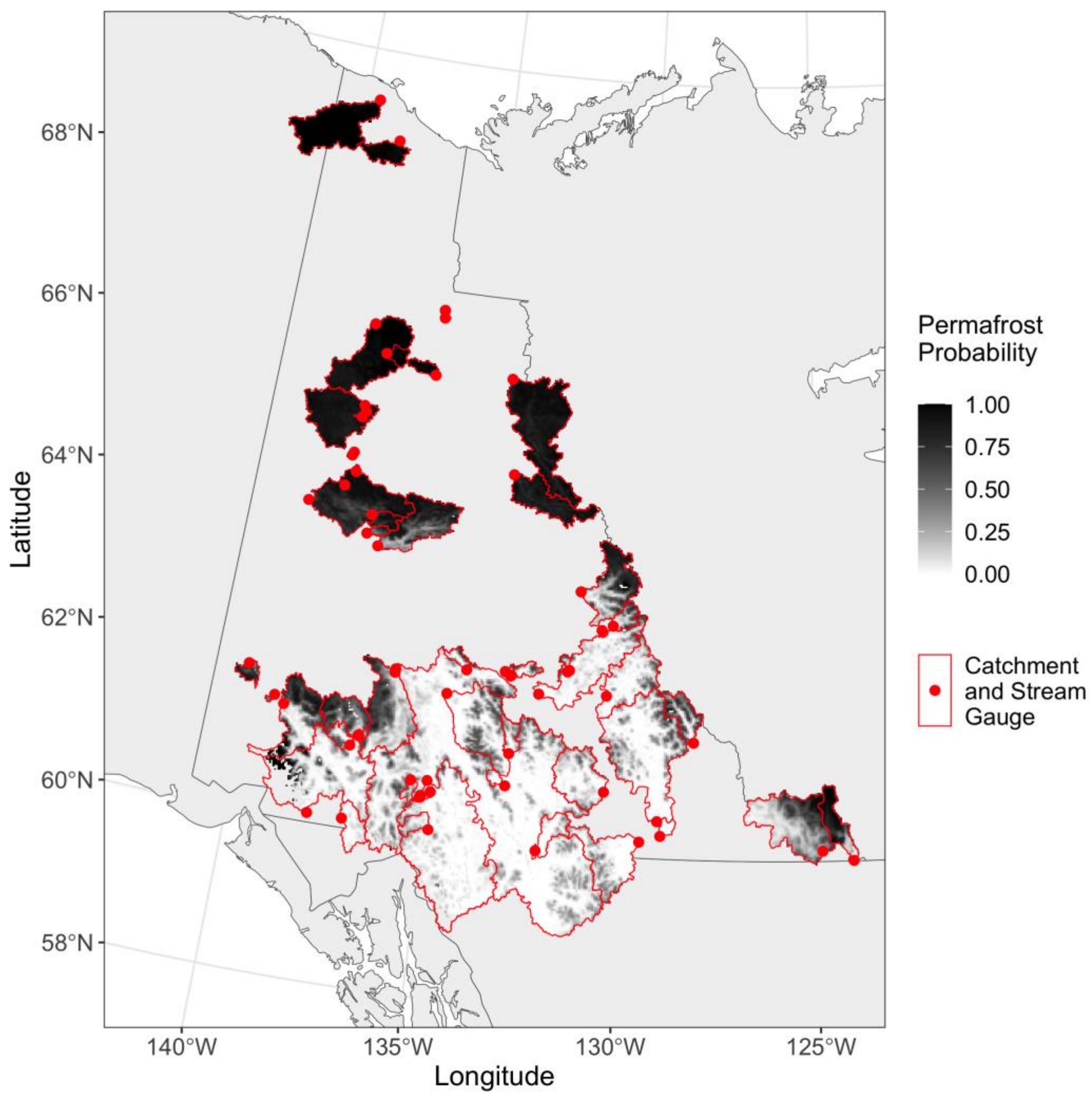

**Figure 1: The 57 study catchments located across Yukon Territory, Canada. Each catchment is filled with the gridded estimated permafrost probability (Obu *et al.*, 2019), at 1 km resolution.**

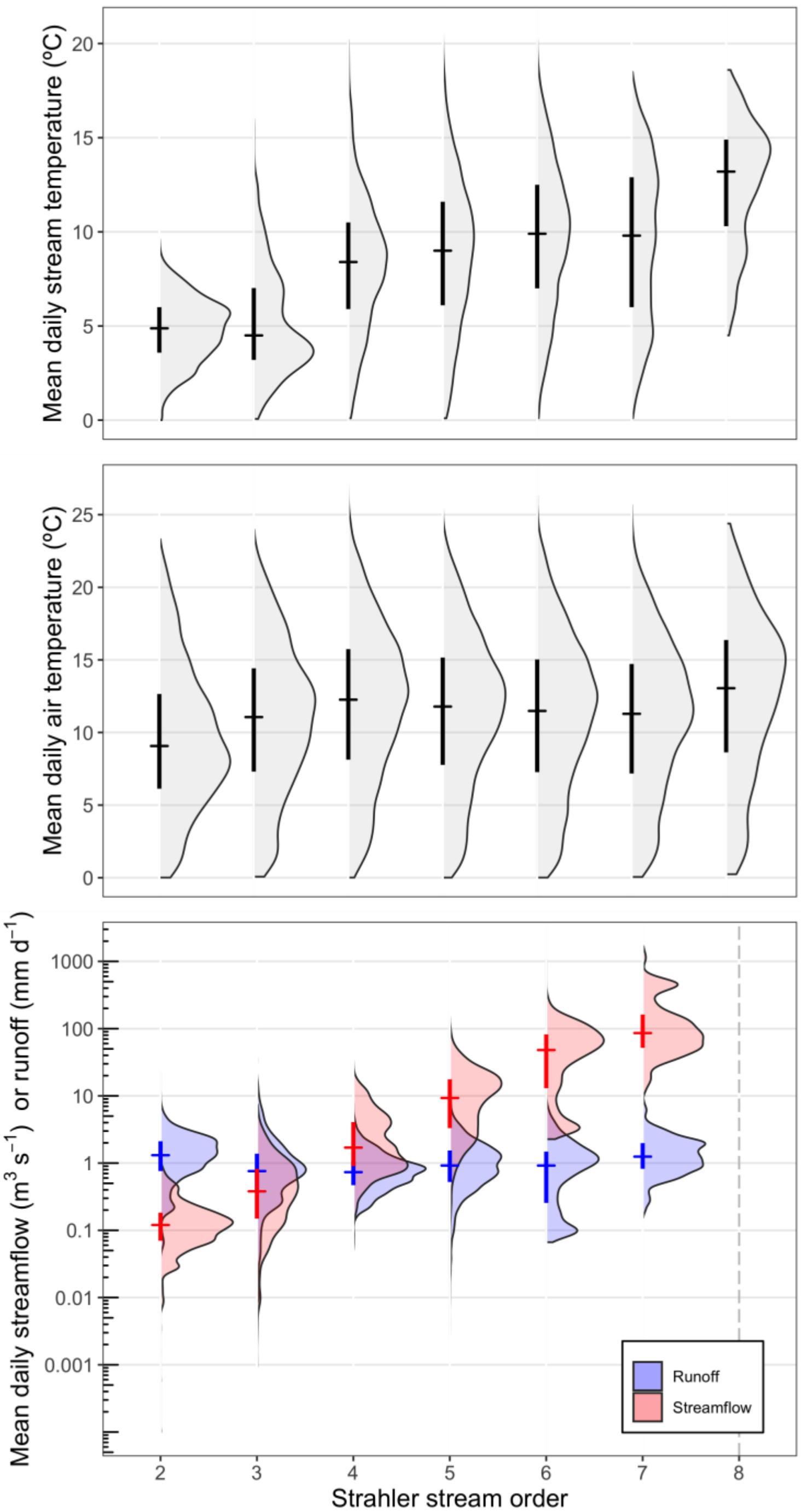

**Figure 2: Distribution of mean daily stream and air temperature, and streamflow by Strahler stream order. The crosses indicate the median and 25th to 75th quantile values range. There is no streamflow data available for the study stream with Strahler order 8.**

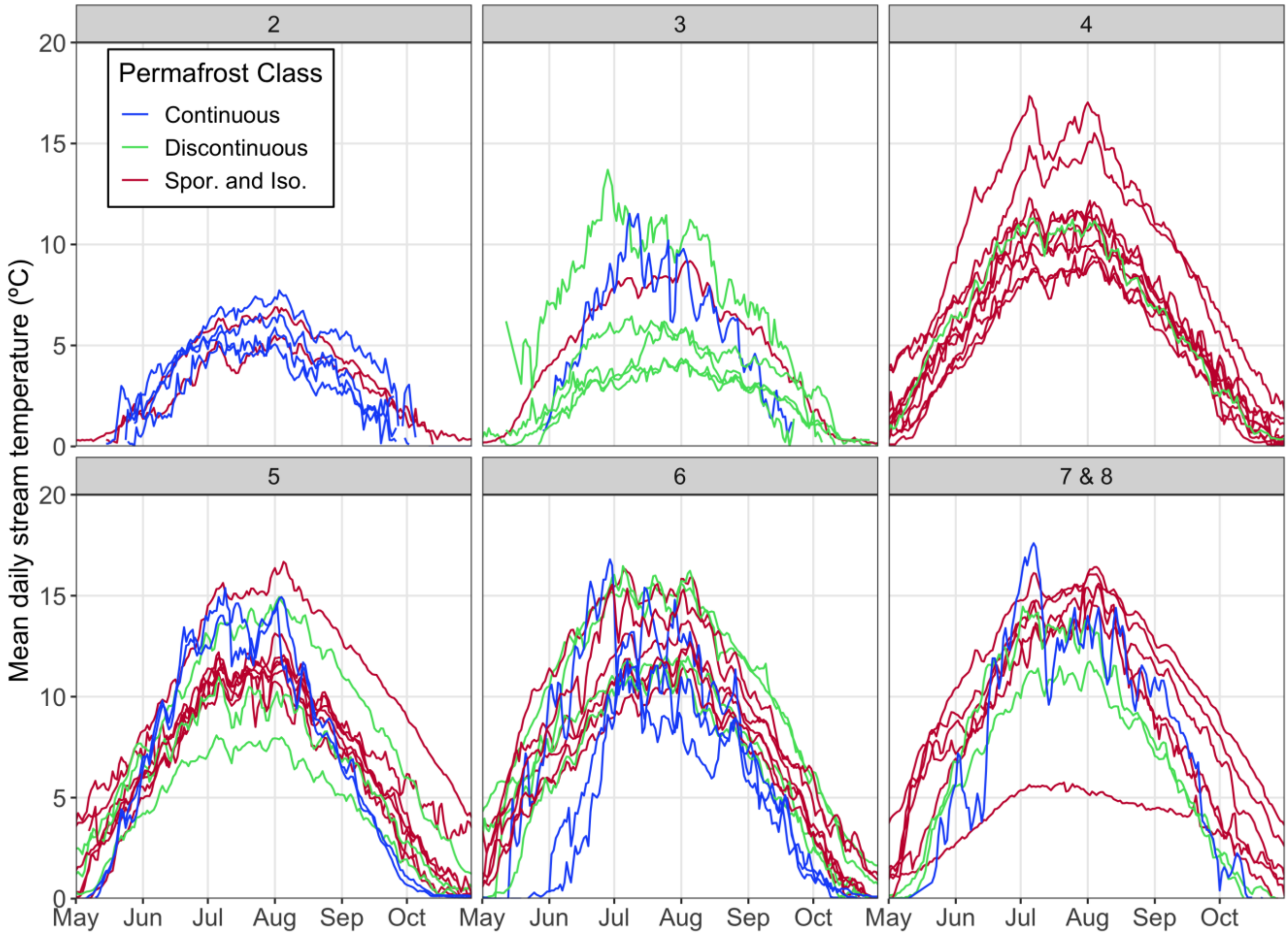

**Figure 3: Thermograph of long-term mean daily stream temperatures for the study streams, with panels arranged by Strahler stream order. The period of data included in the analysis for this study is from 1 July to 15 October.**

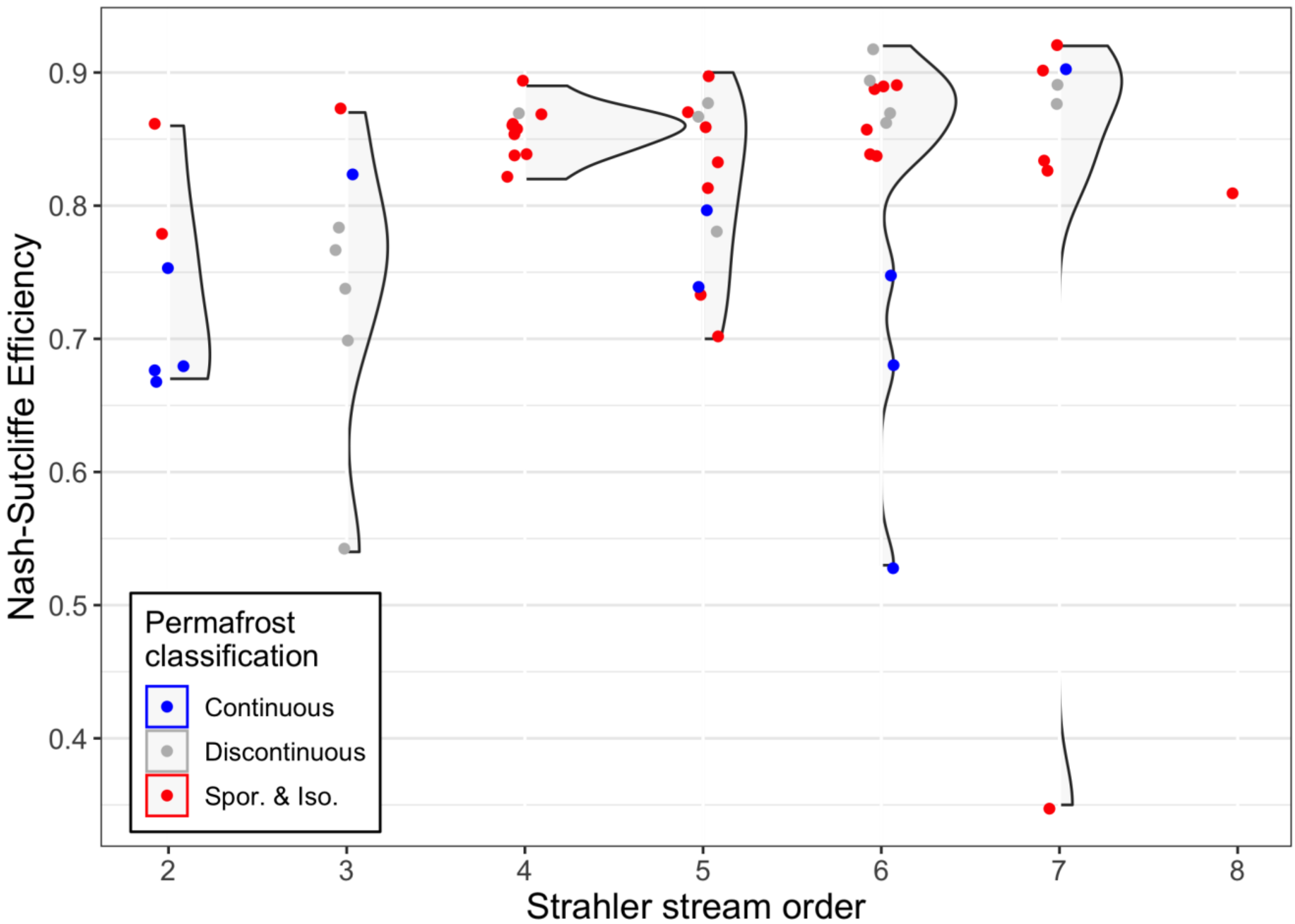

**Figure 4: Distributions of the Nash-Sutcliffe efficiencies of the linear models relating stream and air temperature per site. The points are coloured according to catchment permafrost classification and are grouped by Strahler stream order.**

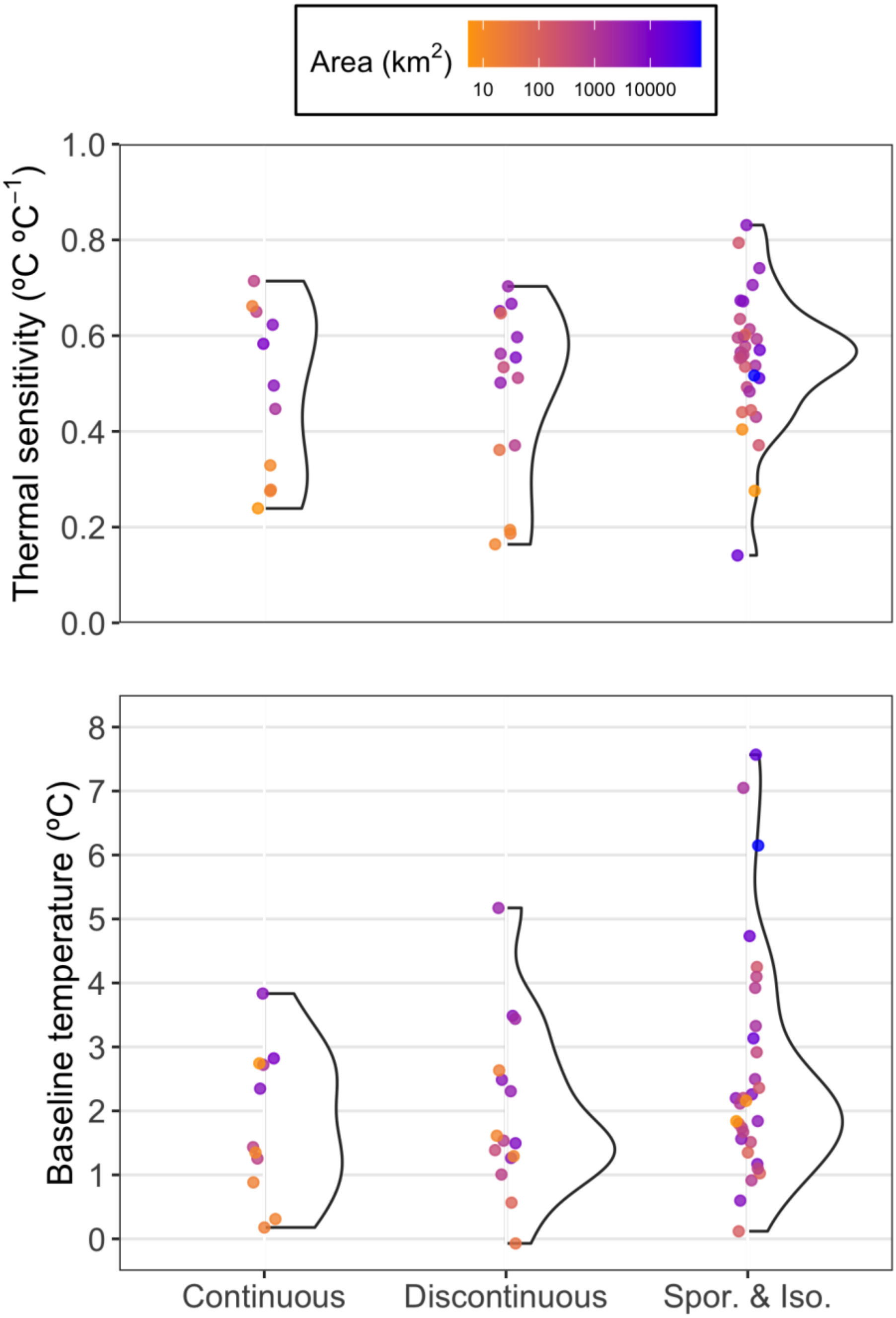

**Figure 5: The distribution of estimated coefficients for a linear relationship between stream and air temperature. The slope is defined as the thermal sensitivity, and the regression intercept is defined as the baseline temperature. The distributions are grouped according to catchment permafrost classification; continuous, discontinuous, and sporadic and isolated.**

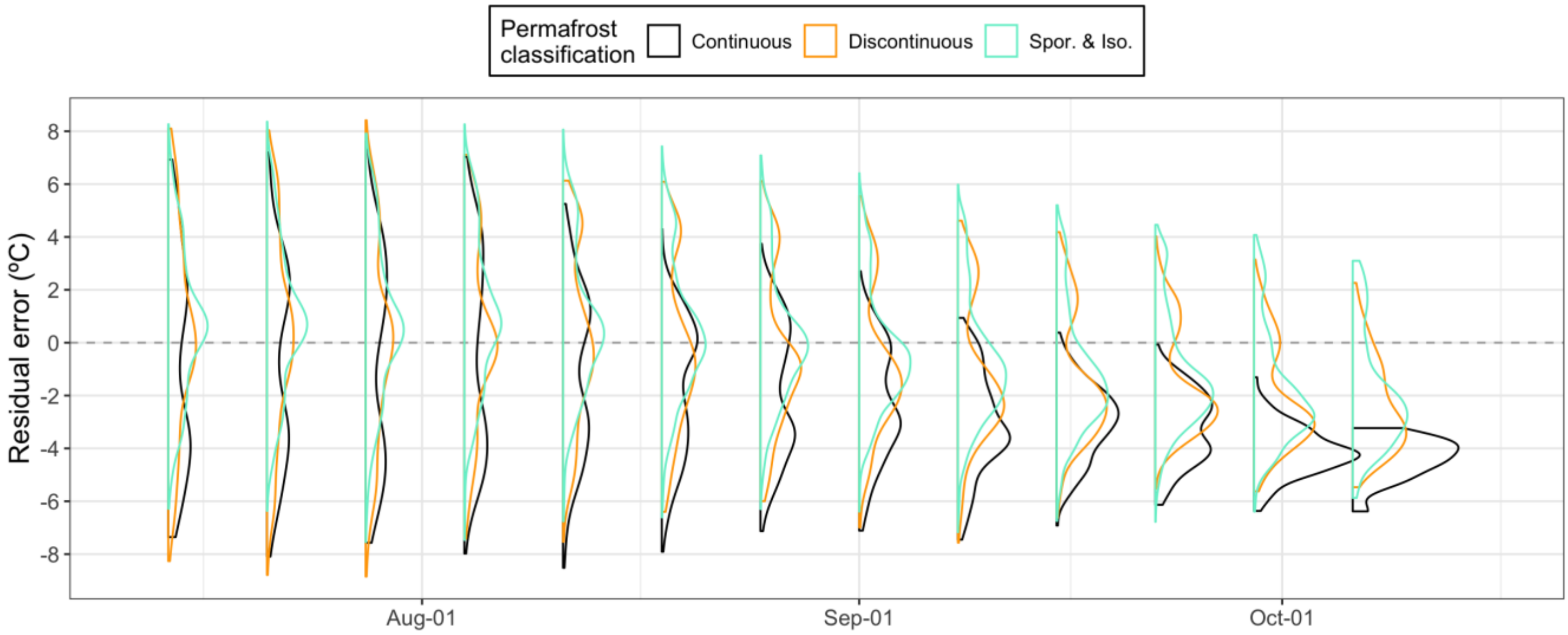

**Figure 6: The weekly residual error distribution as estimated by applying a linear regression fitted to the first two weeks of July. The residuals displayed correspond to dates on or after 15 July, and the distributions are grouped by permafrost classification.**

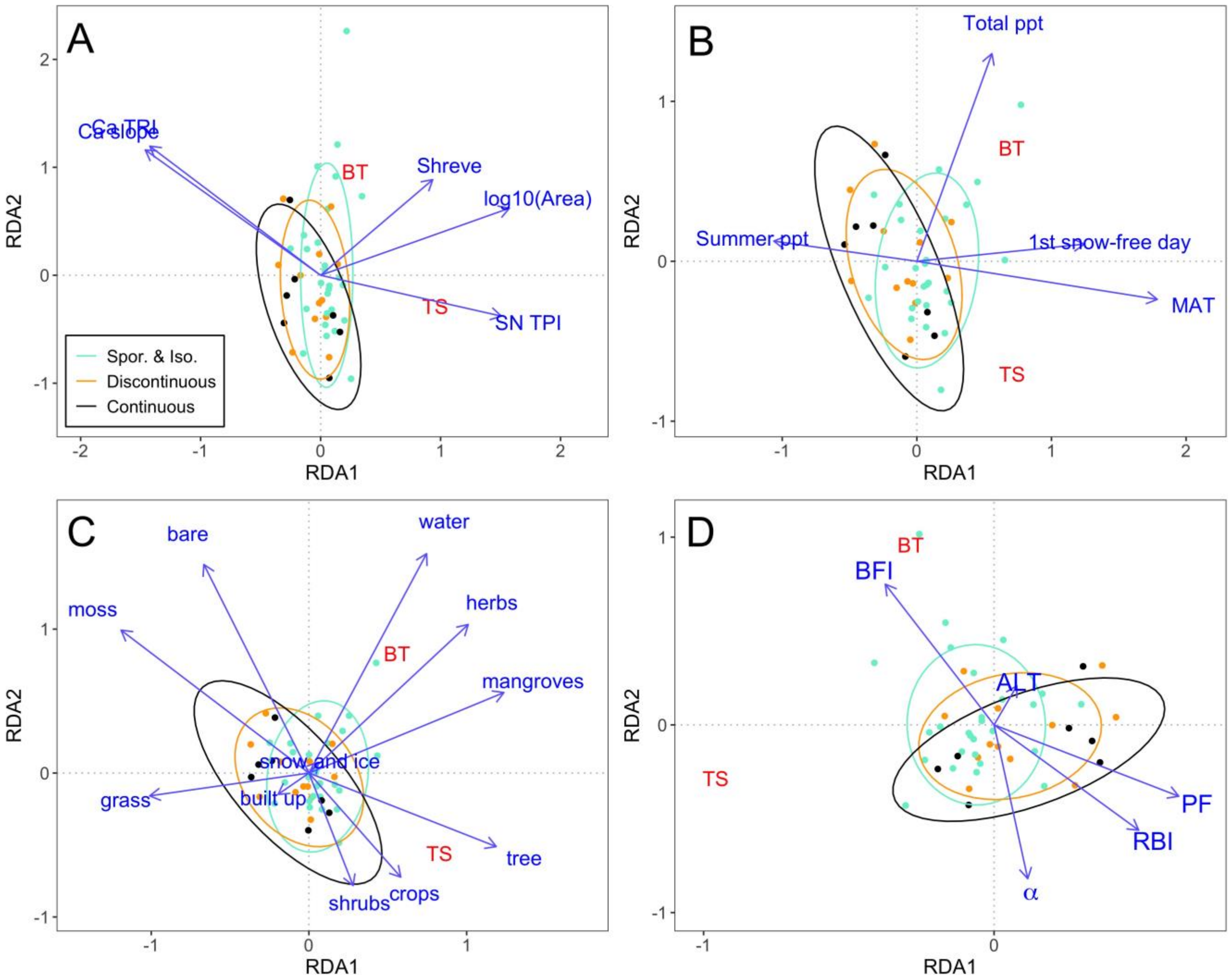

**Figure 7: Redundancy analysis between thermal sensitivity (TS) and regression intercept (BT) and four groups of characteristics and variables corresponding to: (a) catchment physiography, (b) climate, (c) land cover, and (d) permafrost presence indicators. Definitions of abbreviated variable names can be found in Table 1.**

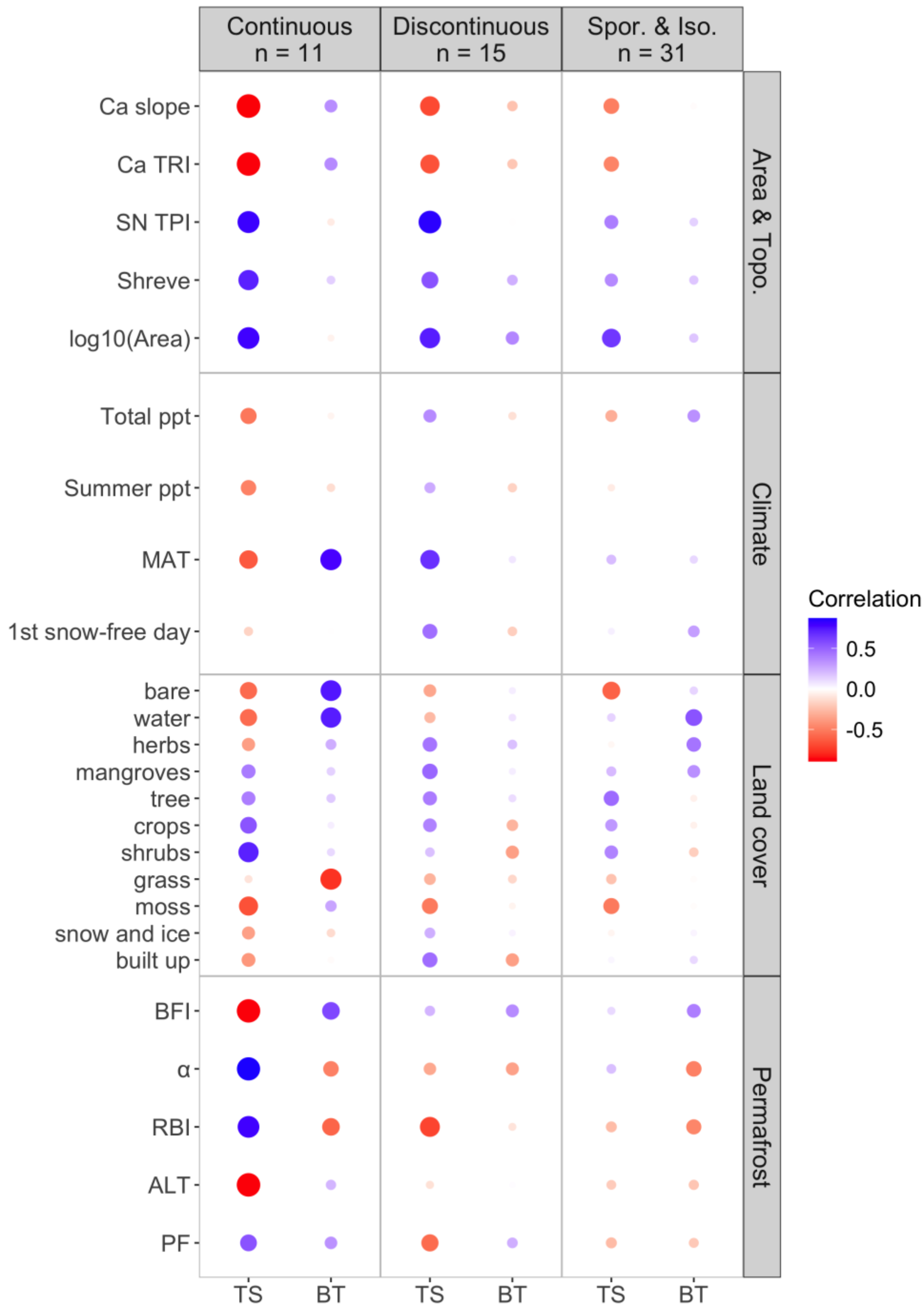

**Figure 8: Pearson correlation coefficients between thermal sensitiivty (TS) and baseline temperature (BT), and the catchment and environmental variables. The correlations were computed per catchment permafrost classification (i.e., continuous, discontinuous, and sporadic and isolated permafrost). Definitions of abbreviated variable names can be found in Table 1.**

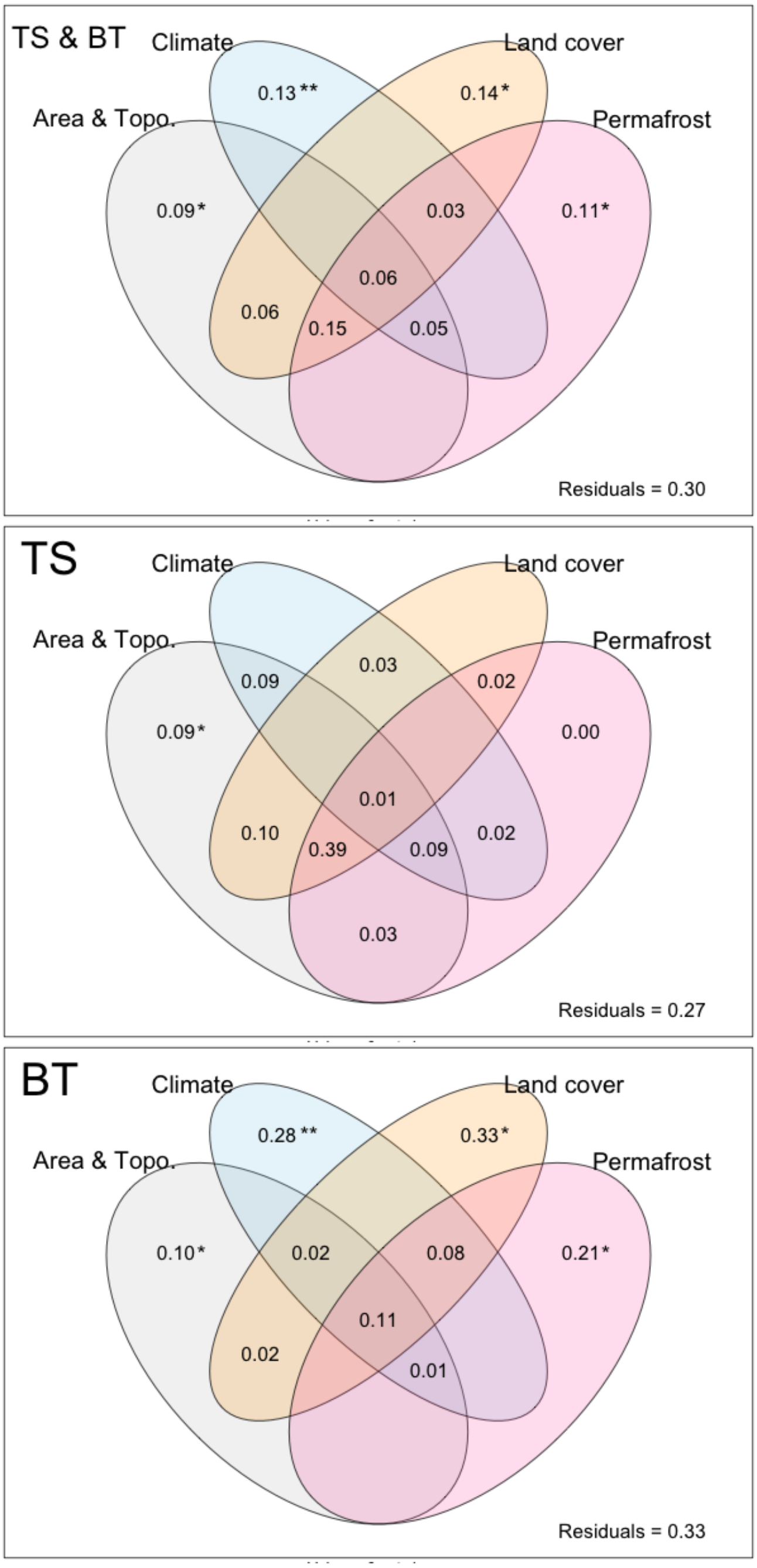

**Figure 9: Variance partitioning of the redundancy analysis (RDA) models fit to four groups of catchment and environmental predictor variables. The values represent the proportion of variance explained by an individual group of variables, or combinations thereof, relative to a global RDA model including all explanatory variables. The panels are labelled according to the response variables included in the RDA models on which variance partitioning was conducted. Panel (a) represents thermal sensitivity (TS) and baseline temperature (BT), panel (b) represents TS, and panel (c) represents BT. Statistical significance of the individual components are indicated with asterisks as follows: * $p < 0.05$; ** $p < 0.01$, and negative values are omitted.**

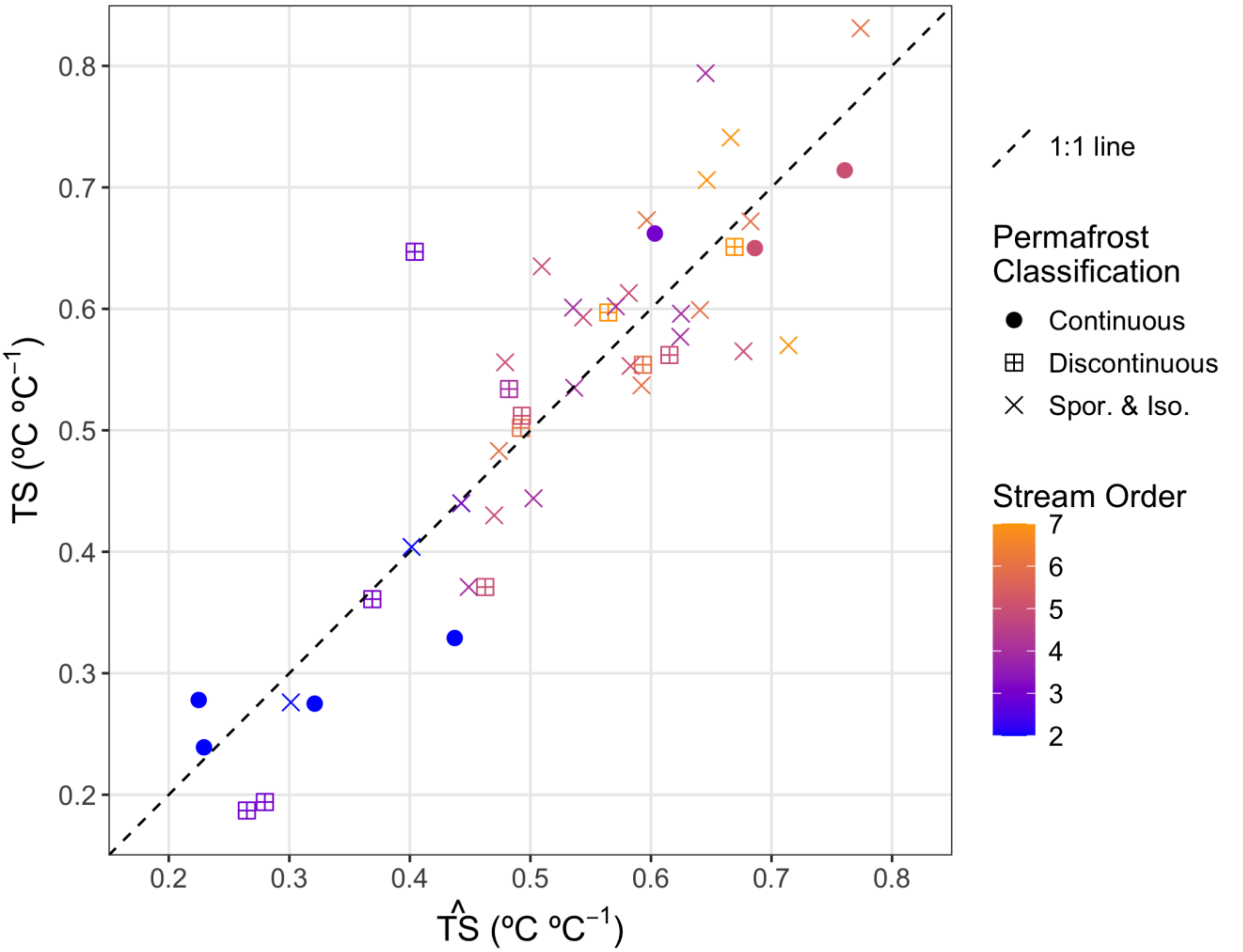

**Figure 10: Catchment thermal sensitivity from the linear regression compared to thermal sensitivity as predicted by catchment properties ($\widehat{TS}$) through multiple regression.**