# Peer review of "The influence of permafrost and other environmental factors on stream thermal sensitivity across Yukon, Canada"

_EGUsphere, 2024_

## Referee Comment (RC2)

[Figure]

[Figure]

[Figure]

Figure 1: Slope of mean daily air temperature vs mean daily water temperature plotted by year by site. Blue and red indicate that rates of decrease or increase are statistically significantly different from zero, respectively (p <= 0.1), respectively. Grey indicates not statistically significant rate of change.

[referee-annotated manuscript omitted]

---

## Author Comment (AC1)

**Replies to Reviewer #1**

We thank the reviewer for their comments and suggestions. Their comments regarding figure captions and interpretation are particularly helpful. Our replies to specific comments are provided below in blue text.

**Specific Comments**

Line 426-437: Why does your model (and other models) over-predict during August – October? Any hypotheses as to why there is this seasonal variability? What is it about the permafrost catchments that makes their deviation over time greater?

We hypothesize the dynamic nature of the active layer, as it increases in thickness through a summer, is a large driver in the increasing model error during August-October due to the shifting catchment soil storage and subsurface runoff processes. We have added to the text to clarify this point.

Figures: There are 10 figures. Could some figures be consolidated/combined or placed in a supplement?

We believe that all of the figures provide valuable context or representation of the data to assist the reader in interpreting the analysis and results. It would be difficult to combine any of the current figures without adding difficulty or confusion in interpretation.

Figure 7 is a hard to interpret. Please provide a description either in the caption or in the text to help the reader interpret.

Additional text has been added to the figure caption to assist the reader in its interpretation.

Figure 9 is also hard to interpret. Could you provide an explanation of how to interpret? Why do some of the regions in the diagram not have a value? Because they are negative?

Some additional text providing examples has been added to the figure caption. The last sentence of the caption specifies that negative values are omitted from the diagram.

---

## Author Comment (AC2)

**Replies to Reviewer #2**

We thank the reviewer for their comprehensive comments and suggestions. Their comments regarding uncertainty assessment and the non-stationary nature of the hydrology in the study region are greatly appreciated and have distinctly improved the clarity and interpretation of the manuscript. Additionally, their detailed technical comments and suggestions for improvement on clarity throughout are particularly helpful. Our replies to specific comments are provided below in blue text, and replies to specific technical corrections are provided as comment replies in the attached PDF.

**Specific Comments**

I provide numerous suggestions for potential ways to strengthen the manuscript in the attached pdf for the authors to consider and highlight two comments below.

First, I suggest a discussion of sources of uncertainty and their implications on the outcome be added to the manuscript. This includes the use of modeled air temperature, the reporting uncertainty of water temperature observations on the order of 0.4 degrees C, etc.

We have conducted an uncertainty assessment on the results of the linear modelling using the assumed error of +/- 0.44ºC and added text and figures (to an SI) to the manuscript to address this point. The model statistics after adjusting the observed stream temperature to incorporate the assumed uncertainty were worse but fairly similar to the model statistics for model using unadjusted stream temperature (e.g., NSE of 0.78 vs 0.81, RMSE of 1.29ºC vs. 1.21ºC, respectively). As such, we suggest it is reasonable to consider the trends and patterns in the modelled output to be representative of reality. We have added text to the Methods and Discussion to incorporate this additional work and commentary.

Second, we know that this system is dynamic and in flux. As such, the thermal processes that drive correlations between air and water temperature are likely also changing over time. I understand the need to use static estimates for active layer thickness and permafrost coverage (limited by data), but that does assume static conditions (over 20+ years in some cases). This should be acknowledged in the discussion and the implications explained. The more interesting/concerning pattern that emerges from the data (which the authors are commended for publishing https://zenodo.org/records/11668943) is that thermal sensitivity varies substantially between years with a median interannual range (max TS – min TS) of 0.22, or 41% of the magnitude of the median TS value. To make it even more interesting, the interannual variability appears to be serially correlated (e.g., demonstrating a temporal trend) with 10 stations decreasing the TS over time and 2 increasing with a p value of 0.1 (see attached figure on the last page of "egusphere-2024-1741_review_and_figure.pdf"). The authors need not add this analysis to their work to make a sound contribution, but the potential (and apparent reality) on non-stationarity should be acknowledged and discussed.

An assessment of temporal trends in thermal sensitivity and thermal regimes across Yukon was the initial research objective I pursued for this study, which forms the basis for the first chapter of my PhD thesis. After some initial analysis, I deemed there to be a lack of sufficiently longterm data at enough study streams to represent the range of catchment sizes considered and across the range of permafrost extents. That said, the figure generated by the reviewer provides compelling evidence that there are some trends in annual thermal sensitivity that should be investigated. We have added a segment to the discussion to acknowledge the topic of non-stationarity and interannual variability in thermal sensitivity associated with changing permafrost. We thank the reviewer for this very insightful comment.

**Technical Corrections:**

See attached document (egusphere-2024-1741_review_and_figure.pdf) for specific technical comments.

Specific replies to the technical comments are provided as replies to comment 'notes' made by the reviewer in the document 'egusphere-2024-1741_review_and_figure.pdf'. The copy with our replies are attached here (egusphere-2024-1741_review_and_figure_Response.pdf).

---

## Author Comment (AC3)

[revised manuscript text omitted]